# Primary Teachers' Challenges in Implementing ICT in Science, Technology, Engineering, and Mathematics (STEM) in the Post-Pandemic Era in Uganda

**Israel Kibirige** 

Department of Mathematics, Science & Technology Education, University of Limpopo, P/Bag X1106, Sovenga 0727, South Africa; israel.kibirige@ul.ac.za

**Abstract:** Information and communication, technology (ICT) has become a necessary tool in education post-COVID-19 pandemic. However, integrating ICT into teaching and learning has been a daunting challenge in many developing countries, such as Uganda. This qualitative case study investigated the challenges that primary school teachers face in implementing ICT in teaching science, technology, engineering, and mathematics (STEM) subjects in Uganda. The study found various challenges divided into three categories: infrastructure and internet connectivity; individual factors and administrative support; and curriculum and learner support materials. A significant obstacle was the lack of computer access, internet connectivity, and ICT textbooks. Additionally, teachers lacked the ICT skills necessary to integrate technology into their teaching methods, and administrative support was insufficient. The identified challenges in this study will require a multi-pronged approach that includes increasing investment in infrastructure, providing training and support to teachers, and developing relevant and appropriate ICT curriculum materials. In conclusion, this study highlights the challenges that primary school teachers face in implementing ICT in teaching STEM subjects in Uganda. By addressing the identified challenges, policymakers and stakeholders can take steps towards improving ICT integration in primary school education and bridging the digital divide in Uganda and other developing countries.

**Keywords:** challenges; digital divide; ICT integration; primary schools; STEM subjects; post-COVID-19

## 1. Introduction

Information and communication technology (ICT) is essential in education, particularly in the post-COVID-19 era. While ICT use in education was encouraged before COVID-19, it became a more necessary mode of teaching following the COVID-19 outbreak, which disrupted the global education system. The need for social distancing led to mobility restrictions for teachers, parents, and learners to the point of total lockdown of the education sector, among others. ICT usefulness in everyday activities [1] was amplified during COVID-19. Khan [2] contends that integrating ICT in teaching and learning in developing countries improves learner performance. Studies indicate that integrating ICT in the teaching of science, technology, engineering and mathematics (STEM) subjects is particularly useful, as it can help learners to understand seemingly abstract science concepts; this assists in cooperative learning to develop the creativity [3] and higher-order thinking processes [4]. Thus, integrating ICT in teaching primary school subjects is vital to meet the challenges of the 21st century [5], and to develop students' proficiencies [6,7]. According to Afutor [8], technological integration is influenced largely by ICT infrastructure and ICT skills, while Rabah et al. [9] emphasize the role of the context. This suggests that specific contexts significantly affect ICT integration in various activities. Rabah et al. [9] further showed that ICT integration should take cognisance of contexts, such as perceptions, infrastructure, and their implications [10]. Attending to these parameters guarantees Africa to meet the United

Nations Sustainable Development Goal four of inclusive quality education for everyone through ICT.

Altınay-Gazi and Altınay-Aksal [11] and Tarus, et al. [12] contend that ICT, videos, and multimedia computer software actively captivate learners in learning. Thus, integrating ICT in teaching primary school subjects is vital to meet the challenges of the post-pandemic era, and to develop learners' proficiencies [6,7]. When ICT tools are used, learners' motivation to learn is increased [13], which increases their conceptual understanding and retention of knowledge [14]. Altınay-Gazi and Altınay-Aksal [11] showed that interactive radios using images and sound to dramatize concepts encourage learners to play an active role in learning. Using ICT tools helps to transmit higher-order thinking skills and creativity [11].

While Bowman et al. [4] show that teachers' perceptions of ICT are linked to their decisions to integrate technology in teaching, Scherer et al. [15] suggest that teachers' intentions to use ICT had no impact on technology use. For over two decades, new technology has been viewed as suitable for the 21st century [16]. Furthermore, it has been predicted that ICT use in primary schools would provide a bright future and lay a solid foundation for the labour market. Despite these benefits, the challenges of ICT use are exhibited in many ways, particularly in the digital divide among developed and developing nations [17], which complicates the incorporation of ICT in primary school education in developing countries [18]. ICT is a basic component for improving thinking, performance, and quality of learning [19]. Thus, integrating ICT in primary school subjects is important in preparing learners, particularly in developing countries such as Uganda.

This study was guided by two theories: technological pedagogic content knowledge (TPACK) [20]; and the digital divide theory [21]. TPACK connects teachers' content knowledge, pedagogy, and technology, which should be integrated into the teaching process [22,23]. TPACK assumes that the teacher should know what to teach and how to teach it and can select an appropriate technology to support lesson delivery. The TPACK framework includes three interconnected elements: content knowledge, pedagogical knowledge, and technological knowledge [24]. TPACK was deemed relevant as it highlights the benefits of using technology in teaching. In addition, the digital divide theory [21] was used. The digital divide theory highlights the digital inequalities that may exist among people in different contexts [25]. This theory was considered appropriate because the study investigated ICT use in two different contexts: urban and rural schools.

Schools are intended to build learners' knowledge and skills. There are two types of schools in Uganda: public and private, where private secondary schools account for 66% of secondary school learners' enrolment [26] and 23% of the total primary school learner enrolment, which is above the world average of 19% in 2020 [27]. Despite the low percentage of private primary school enrollment, private primary schools play a vital role in education. In Uganda, like in Kenya, most studies have focused on public secondary schools [28,29], and more studies have investigated ICT use in secondary and tertiary education institutions. Apart from Atakorah et al. [30], who studied the challenges of ICT in the Seventh Day Adventist (SDA) College of Education in Ghana, there are no studies regarding the challenges of integrating ICT in teaching in private SDA primary schools. This study is the first to document the challenges of ICT integration in teaching STEM in private SDA primary schools in Uganda. Therefore, this study explored teachers' challenges in integrating ICT in teaching STEM subjects in two private SDA primary schools in rural and urban contexts. The study answered four questions: (1) What ICT do you use in teaching and for management?; (2) How does the school support teachers to integrate ICT in teaching STEM subjects?; (3) What challenges do teachers face in integrating ICT in primary school STEM subjects?; and (4) How do you overcome those challenges in lesson preparation and teaching in the classroom?

## 2. Materials and Methods

A qualitative approach [31,32] was used because of its flexibility [33], where the number of participants can be few or many depending on the context. Also, qualitative

studies offer in-depth information that quantitative studies cannot achieve. A case study design [31] was employed to explore teachers' challenges in implementing ICT in primary schools in Uganda. This case study explored teachers' challenges in integrating ICT in teaching STEM subjects in primary schools in Uganda. A researcher-designed interview guide with follow-up questions was used to collect data during post-COVID-19: January–February 2023. The target population comprised two schools: one from an urban context (School U) and one from a rural context (School R). The Urban school is in the Wakiso district, and the Rural school is in the Kassanda district in Uganda. These schools were [34] selected because both are primary schools and represent urban and rural contexts, respectively.

Both schools are mixed gender. The researcher used a purposive sampling technique to select four participants: a deputy headteacher and a teacher from the urban school and a headteacher and a teacher from the rural school. They were chosen because they had experience in primary school contexts. The sample comprised four male participants because they taught STEM subjects. According to Creswell and Creswell [31], a sample of 2 to 25 participants is sufficient in a qualitative study. This sample was considered to have the required information regarding the challenges of ICT use in primary schools. The teachers' teaching experiences ranged from 1 year to 34 years. The deputy headteacher from the urban school had a bachelor's degree and a master's degree in teaching, and the teacher had a primary certificate, a diploma in education, and a bachelor's degree in teaching. From the rural school, the headteacher had a diploma and a bachelor's degree, and the teacher had a bachelor's degree in teaching. The participating schools are codenamed U and R for the urban and rural schools, respectively, for confidentiality. The participants are TU1 and HU2 for the teacher and deputy headteacher from the urban context, and TR3 and HR4 for the teacher and headteacher from the rural school, respectively.

The researcher designed four interview questions (Appendix A). Two experts, a professor and a lecturer in education, validated the questions. The recommended suggestions from the two experts were implemented before data collection. The open-ended interview questions included: what ICT was available in the schools, how the school supports ICT integration, what challenges were experienced in integrating ICT, and what strategies were used to ameliorate the challenges. The researcher used start-up and rounding questions [35] like: "Can you tell me about the challenges you experience in integrating ICT..." and various follow-up questions (Appendix A). The researcher conducted individual interviews with the four participants from the schools. Qualitative interviews were used because they provide insights according to the participants' views. Thus, the data collected reflected the participants' views, experiences, and meanings attached to different issues [36]. Also, interviews provide unique data, which cannot be found using other data collection methods [37]. All interviews were audio recorded for 30 to 40 min.

*Data Analysis*

The data were transcribed and analysed thematically using open, axial, and selective coding [38]. In open coding, sentences were read line by line to decipher the main concepts. The researcher used three steps of axial coding: inductive and deductive analysis, focusing on theory construction and determining the truth from the data, respectively; making constant comparisons of data, sub-themes, and codes generated; and reading line by line and keeping focused and engaged to avoid bias [39] (Figure 1). Furthermore, the researcher considered the causes, context, and conditions in which the study was carried out. During selective coding, sub-themes were merged to form the main themes [40] concerning the study.

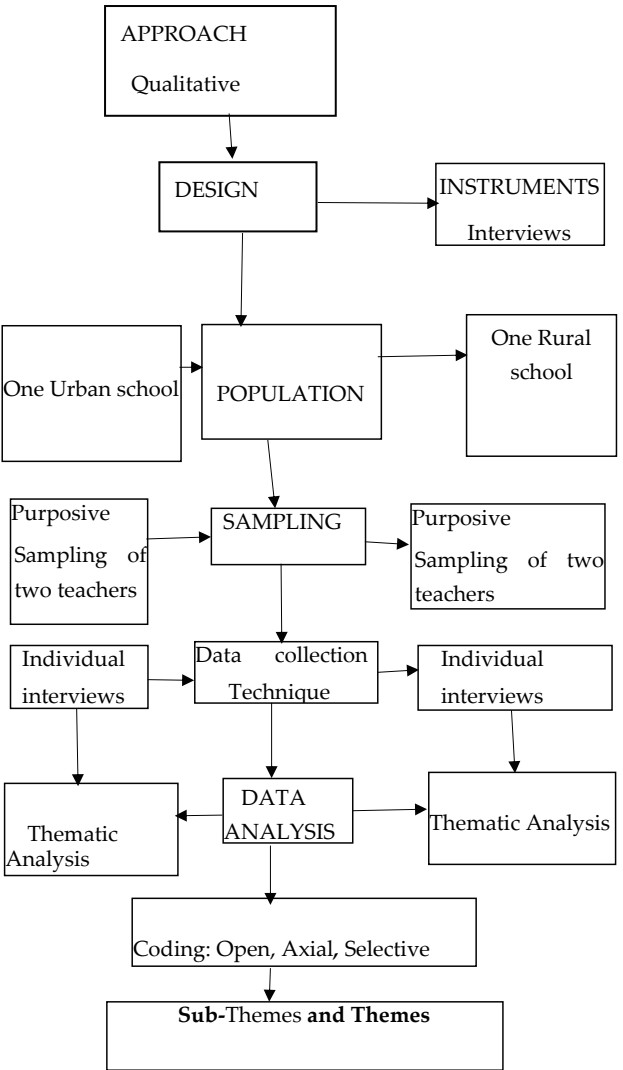

**Figure 1.** A methodological Scheme showing the process followed to generate sub-themes and themes.

### 3. Results

The results show challenges in integrating ICT in teaching STEM subjects in primary schools in Uganda. Using thematic analysis, three themes were identified: (1) Infrastructure and internet connectivity; (2) Individual factors and administrative support; and (3) Curriculum and learner support materials (Table 1).

Participants explained the integration of ICT in teaching and for management in the following extracts.

Participants TU1 and HU2 agreed that the school uses computers, data projectors, the internet, email, and smartphones to teach STEM subjects:

> "Teachers and learners access the internet at the laboratory using the personal password generated for each user. The school uses an electronic register system where registration takes place; parents use e-registration to pay the school fees and other charges. . . . The school uses a learning management system". (TU1).

In addition, participant HU2 stated:

> "We use phones, email, and WhatsApp to communicate with learners, parents, and the public. Also, we use radio and videos, YouTube and some teachers use DVDs to show images of various objects and structures to enhance learning". (HU2).

The teacher and the headteacher from the rural school concurred that the school uses one tablet and smartphones. The school has one tablet, and the school has an email address

that is accessible to teachers. The teachers were not sure whether any learner had an email address. The following excerpts illustrate the participants' sentiments:

"The school has only one tablet, which teachers use to demonstrate various images or difficult concepts. Teachers use their smartphones to search for online information for edifying lessons and communication. Aahh!... I have not seen any learner here with a smartphone; owning a smartphone and accessing data is beyond our learners' income". (TR3 and HR4).

**Table 1.** A schematic representation showing how the axial coding process was conducted.

| Context | OPEN | AXIAL | SELECTIVE | THEME |
|---------|------|-------|-----------|-------|
| Urban | Few computer laboratories/ ICT buildings Few desktop computer installations Data projector DVD No laptops | Laboratory Computers, buildings Computers | Hardware | Infrastructure and internet connectivity |
| | Communicate using smart phones Poor internet and Wi-Fi No computers No laptops No data projector One tablet | Cell phones Connectivity | Connectivity | |
| Rural | No laboratories, no computer buildings to access ICT No access to electricity at school | | Hardware | |
| | Only teachers communicate using smart phones Poor internet and Wi-Fi | | Connectivity | |
| Urban | Internet data paid by the school Teachers' attitudes towards ICT Teachers advised to attend workshops twice a week to gain ICT skills Tired of attending workshops Some teachers miss attending ICT workshops ICT Skills | Payments Practice using ICT Training | School management Personal | Individual factors and administrative support |
| Rural | Teachers' attitudes towards ICT Teachers access electricity outside campus Teachers and learners share one tablet | Teachers' ICT integration Teachers' proactivity Cooperative group work | Personal | |
| Urban | No ICT content materials for primary school Schedules for using ICT as a subject and as integration into teaching content | Learners' materials ICT content integration | Learner support Materials Curriculum | Curriculum and learner support materials |
| Rural | Inability to access videos for teaching challenging topics Time and periods for ICT Lack of models of ICT integration in class Computer learning materials such as ICT textbooks, DVDs, not available to teachers | Lesson planning Lack of resources | Learner support materials Curriculum | |

### 3.1. Theme 1: Infrastructure and Internet Connectivity

Teachers from both urban and rural schools experienced structural and connectivity challenges in integrating ICT into the teaching of STEM subjects. The teachers from the urban school indicated the high cost of data as the main challenge, which limited ICT use for learning as shown in the extract below.

"The data the school buys intended for a month can get finished in two to three weeks. In order to curb data expenses, the internet is switched on at particular times. Some of the excessive data usage is due to learners using computers for other websites other than those specified for learning specific content. This challenge may have emanated from the COVID-19 lockdown, where learners were using ICT with no one to limit their use. Thus, we are challenged to manage learners on what to stick to for learning purposes". (TU1).

"There needs to be more computers to cater for all learners in the class. The problem is compounded by a lack of technicians to repair the broken-down computers in the required time frame". (HU2).

"The curriculum requires a computer, laptop, or smartphone per learner, which is impossible due to financial constraints. In our school, three learners share one computer; the learner-to-computer ratio is 3:1". (HU2).

In the rural school the teacher stated that:

"Our school does not have computers. We have one tablet, which we share among ourselves. Even if it had them, the school could not afford the high cost of the data. To make matters worse, the school is neither connected to electricity nor has a solar system. Data is also expensive. In addition, most of our teachers have not been exposed to ICT. We are seeking donations to increase the number of tablets". (TR3).

### 3.2. Theme 2: Teacher Factors and Administrative Support

The school supports individual learners and teachers to use ICT in education. The school supports teachers in integrating ICT into teaching through professional development programmes. Excerpts from teachers are used as evidence:

"There is also a computer for the staff dedicated to research when teachers are making lesson plans". (TU1 and HU2).

When teachers were asked whether there were any teacher training programmes, they responded as follows:

"Yes, the school provides training twice a week. Each session lasts two hours. Ahhh, the new teachers are trained on Sundays to avoid clashes among the teachers because those teachers are, at times, starting from the basics, yet the old teachers are at advanced levels in integrating ICT into teaching. Teachers are supported by the school, which has dedicated a computer and data projector to use in class for learners to view various images, objects, and videos to enhance learning". (TU1).

"Some teachers have ignored ICT use in teaching, especially after schools opened for face-to-face teaching, and this has resulted in poor attendance of ICT training. Some teachers lack competencies regarding the integration of ICT in STEM subjects". (HU2).

The headteacher and the teacher from the rural school attested that the school did not offer ICT as an examinable subject, as shown in the excerpts below:

"In our school, there is no subject called ICT. The school did not register learners for a separate course in ICT. The school may offer ICT as a separate and examinable subject in the future. None of us uses blogs, and we do not know much about them". (TR3).

The headteacher from the rural school added:

"Even if the curriculum could allow ICT to be taught in primary schools, our rural schools may not afford to offer it due to lack of computers and electricity supply, leave alone the high cost of data". (HR4).

The head-teacher from the rural school further stated:

"Another thing is the viruses that attack the system regardless of the installed antivirus. Again, computers get stolen or vandalised from schools which have them". (HR4).

### 3.3. Theme 3: Curriculum and Learner Support Materials

The teachers and the deputy headteacher explained the stipulations of the curriculum regarding ICT in primary schools:

"The curriculum stipulates that learners should learn ICT during the two periods of ICT in primary three to seven, which makes some teachers feel exonerated from integrating ICT into the various subjects". (TU1 and HU2).

"To meet the curriculum requirements, teachers sometimes download simulations and save them on DVDs for use in the classroom when the internet is switched off". (HU2).

In addition, the headteacher from the rural school suggested that

"There should be a tax waiver on computers and data projectors for rural schools". (HR4).

"Also, I mean teachers are to be advised on teaching videos and virtual learning materials that can be accessed offline, which should be useful during the electricity blackout. Finally, more signal towers should be installed for accessing the internet". (HR4).

## 4. Discussion

This study investigated primary school teachers' challenges in integrating ICT in teaching STEM subjects in primary schools during the post-COVID-19 era. The results show that urban and rural primary schools experienced challenges in integrating ICT into teaching STEM subjects. The challenges were presented in three themes or categories: (1) infrastructure and connectivity; (2) teacher factors and administrative support; and (3) curriculum and Learner Support Materials (LMS). Also, the strategies used to address the various challenges, the study contribution, and recommendations for improving ICT integration in the learning and teaching of STEM subjects in primary schools are discussed.

### 4.1. Infrastructure and Connectivity

Teachers from the urban school indicated that they had access to computers and other technologies to use in the teaching of science subjects, while the rural school only had one tablet, and the school did not have access to electricity. This disparity in access to infrastructure means that teachers from the urban school had opportunities to enrich their lesson planning and delivery using different technological tools, which are likely to improve learners' conceptual understanding. The lopsided access to technological tools between rural and urban schools is representative of the rural-urban digital divide [25]. The lack of basic infrastructure in rural schools exacerbates learners' vulnerability to poor performance and social exclusion [8]. This inadequate access to technology is a great disservice to the rural school, where not every learner may use ICT for effective learning. A study by Belay [41] showed that some students encounter internet and electricity for the first time while attending universities, which highlights the severity of the lack of electricity in rural areas. This could explain the lower grades achieved by learners from rural schools when compared to those from urban schools.

Both urban and rural schoolteachers in Uganda experienced infrastructural challenges. These challenges imply that the country needs to invest more in infrastructure to accommodate ICT teaching in primary schools. This observation agrees with Muweesi et al.'s [42] study of five public schools around Kampala, Uganda. They also assert that this infrastructural challenge is common in many Sub-Saharan African countries. It is worth noting that the infrastructure challenge is coupled with large numbers of learners; this increases the learner-to-computer ratio in classes. The large learner-to-computer ratios are especially evident in Zambia, where the ratio was 143:1 [43], whereas in South Africa it was 17:1 [44], and a ratio of as low as 3:1 is reported [45]. While these studies show an oblique picture of Africa, the urban school in this study used a rotational approach where three learners shared one computer, suggesting that the learner-to-computer ratio is 3:1. While this ratio is good in a developing country such as Uganda, the restricted use of computers is yet another challenge. The learner-computer ratio alone may not give an accurate picture of ICT integration in learning [46]. In the rural school looked at in this study, the learner-to-computer ratio is not applicable since they have no computers.

Nevertheless, the cooperative learning approach, where learners share through a rotational approach in their groups, is commendable. While teachers from the urban school indicated that they had access to computers, smartphones, and employed radios, videos, e-mails, and WhatsApp in teaching science subjects, their rural school counterparts only had one tablet, and the school did not have access to electricity. This observation is not surprising; Tshukudu et al. [47] have noted that rural schools lack electricity. Hence, Uganda's Vision 2040 to enhance quality education through technology may not be realised. This state of deprivation of basic infrastructure may be responsible to a great extent for the chronically poor performance in STEM subjects in rural areas. For instance, it seems unlikely that all the teachers and learners can effectively access one tablet in a school. Learners are disadvantaged in accessing ICT and thus in understanding some abstract science concepts, which are otherwise clear when ICT is integrated into the teaching.

Clearly, this shows the digital divide and education inequalities post-pandemic at transnational and intra-national levels. This observation concurs with Huffman [48], who found that the digital divide could exist in the same district. This divide can be solved when governments invest in ICT, from preschools to tertiary institutions.

The teachers from the urban and rural schools complained that data was very expensive, and even the school with the necessary infrastructure switched off data to minimise costs. This means that schools continue to teach without ICT because data is not affordable. If schools cannot afford data to integrate ICT into learning, learners cannot develop the 5th Industrial Revolution (5IR) skills necessary for a human- and technology-based society [49].

*4.2. Teacher Factors and Administrative Support*

The participants in the study were all qualified to teach the respective subjects. However, their competencies to incorporate ICT into teaching needed to be improved. Teachers' low ICT skills are understandable, given that the use of ICT in teaching is a relatively new phenomenon in Uganda. This finding is consistent with Townsend [50], who found that most teachers had challenges integrating ICT into their teaching. ICT use in teaching was popularised during the COVID-19 lockdown when schools in Uganda were closed for almost two years, and teachers were compelled to use technology to enable virtual learning in various disciplines, such as veterinary medicine [51] and education [52]. Teachers from the urban school indicated that their school had a designated capacity-building strategy to develop teachers' ICT skills to use ICT in their lesson preparation and teaching. Teachers from the rural school had minimal exposure to ICT, and a lack of basic ICT infrastructure and electricity compounded the situation. Teachers from the rural school had more ICT competence challenges and minimal administrative support to integrate ICT into their teaching. The teachers' ICT competency challenges corroborate the findings of Albahiri and Alhaj [53], who reported on teachers' low ICT competencies. This difference in teachers' competencies and administrative support will aggravate the digital divide between

rural and urban areas within the same country. This observation is not surprising, since similar findings have been reported in countries where the digital divide was reported between individuals and geographic areas [17,54]. Furthermore, Oduor [28] and Afutor [8] attest that teachers' ICT competencies are critical in integrating ICT in classrooms and for fostering learner motivation and conceptual understanding.

Notwithstanding the capacity-building program implemented in the urban school, resentment towards ICT by some teachers was reported. Studies show that teachers' attitudes affect ICT integration in the classroom [4,55]. The various challenges have great implications for meeting the objectives of the National Curriculum to improve learners' interest and participation in STEM subjects. Using ICT tools helps transmit skills, develops higher-order thinking, and enhances creativity [11].

### 4.3. Curriculum and Learner Support Materials

Coping with the new curriculum requires teachers to incorporate ICT in teaching primary school subjects. However, some teachers believed ICT integration was limited exclusively to the two periods per week allocated to ICT, which discouraged them from integrating ICT into the various subjects. In addition, the headteacher from the rural school indicated that support materials such as videos and simulations that could be used offline would be useful in their under-resourced situations.

The study findings show that while the National Curriculum advocates for incorporating ICT into education, teachers need computers, laptops, or tablets and the competencies to integrate ICT in the classroom. Some teachers in rural areas were not aware of websites which were introduced ten years ago by the Uganda government for teachers to access [56], namely: The connect-ED project, CurriculumNet Project (https://www.idrc.ca/en/article/casting-curriculumnet-wider?PublicationID716, accessed on 20 January 2023), and the VSAT project and SchoolNet Uganda project; Content Development project at National Teacher Colleges; Connecting Classrooms project; UConnect More effort is needed to make primary school teachers aware of the available ICT support materials.

### 4.4. The Study Contribution and Recommendations

This study adds to the literature regarding the challenges of integrating ICT in teaching STEM subjects in primary schools post-COVID-19 era. It also contributes to the cognisance of the role of ICT in teaching STEM subjects in primary schools. The study recommends that the government and education stakeholders focus on funding ICT facilities in primary schools equally to minimize the digital divide among different school categories: government-private, rural-urban, single-gender, and mixed-gender primary schools in Uganda. Furthermore, integrating ICT for STEM subjects in primary schools in Uganda requires a massive teacher professional development programme in ICT.

## 5. Conclusions

This research study aimed to investigate the challenges faced by primary school teachers in integrating information and communication technology (ICT) in teaching science, technology, engineering, and mathematics (STEM) subjects during the post-COVID-19 era. The study focused on two private Seventh-Day Adventist primary schools in Uganda, representing urban and rural contexts. The study explored the following research questions: What ICT do teachers use in teaching and management?; How does the school support teachers to integrate ICT in teaching STEM subjects?; what challenges do teachers face in integrating ICT in primary school STEM subjects; and how do teachers deal with ICT challenges in teaching? The study identified three main challenges: infrastructure and internet connectivity; individual factors and administrative support; and curriculum and learner support materials. The need for adequate infrastructure and internet connectivity was a significant challenge for rural and urban schools. Although the urban school had a 3:1 learner-to-computer ratio, this is unlikely to represent most primary schools in Uganda and other developing countries. The study found that the challenges faced by teachers

in integrating ICT had significant implications for meeting the objectives of the National Curriculum to improve learners' interest and participation in STEM subjects.

**Funding:** This research received no external funding.

**Institutional Review Board Statement:** The study was conducted on two private schools and the ethical issues were catered for by receiving permissions from the Director of education for Central Uganda Conference (CUC) of the Seventh-Day Adventist Church. Also, permissions were granted by the leaders of the two primary schools in Uganda. These three permissions served in lieu of ethical clearances and were submitted to the Journal. Hence, there was no need for ethical clearance.

**Informed Consent Statement:** Informed consent was obtained from all subjects involved in the study. Written informed consent was obtained from the study participants to publish this paper.

**Data Availability Statement:** Data for this research is unavailable for sharing due to privacy restrictions.

**Conflicts of Interest:** The author declare no conflict of interest.

## Appendix A  Interview Questions

Please tell me:

Question 1. What ICT do you use in teaching and for management in your school?
*Follow-up questions*

–　Do you have a school email address for teachers? How do they use email?
–　Do you provide the school email address to learners? How do they use email?
–　As a staff member, how do you use the school email address to effect teaching?
–　Does your school use an electronic register system, and how do you use it?
–　Does your school use any learning management system (LMS)?
–　What other communication tools do you use?

Question 2. How does the school Support teachers in integrating ICT in the classroom?
*Follow-up questions*

–　How does the school ensure teachers are updated with ICT use in STEM subjects?
–　What equipment is available for training teachers, such as whiteboards, laptops, and tablets in your school?
–　Do you use emails, WhatsApp groups, and blogs to enhance your teaching?
–　Is ICT taught as an integrated or separate subject in primary school?

Question 3. What challenges do you face in integrating ICT into teaching?
*Follow-up questions*

–　Can you tell me about the challenges you experience in integrating ICT
–　Describe at least one major challenge you face using ICT in teaching in primary school.
–　How do you use the available ICT in your school?

Question 4. How do you overcome those challenges in lesson preparation and teaching in the classroom?
*Follow-up questions*

–　Any other issues regarding ICT in your school?
–　If the teacher wants to use Wi-Fi using the computer dedicated to the staff, what are the procedures to access it?
–　Does the school have mechanisms to overcome ICT challenges? Mention a few specific examples.

Finally, do you have any additional information regarding ICT integration in teaching STEM subjects in your school?

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
