# Peer review of "Primary Teachers’ Challenges in Implementing ICT in Science, Technology, Engineering, and Mathematics (STEM) in the Post-Pandemic Era in Uganda"

_education, doi:10.3390/educsci13040382_

Round 1
Reviewer 1 Report
This is a very interesting paper about the use of ICT in STEM education in Uganda. The study examines both an urban school and a rural school. This paper has some very interesting findings that require action by policy makers to ensure Ugandan students are well prepared for the future.
The content is described and contextualised well to previous and present research and the references are good. Noting there are a few citation errors in the paper.
The research design is clearly stated, but it lacks research questions, and hypotheses. The method is clearly stated but it would be good to include the survey questions.
The discussion of the findings are coherent, and compelling, though there is a lot of repetition that is adding to the length of the paper.
The conclusions are supported by the results but the conclusion is too brief and this needs to be fixed to ensure the best outcomes from this paper.
There are some trivial mistakes throughout the paper that I have highlighted in the attachment. But just a few trivial things to fix:
1) ICT is not a buzzword
2) Purposively is not a word, use purposely
3) Don't start a sentence with [], eg instead of [2] say Khan [2]
4) The phrase 21st century is over used
5) Use commas when referring to multiple sources eg [8], [9]
6) Remove ICT gadgets, just using ICT is enough - you don't need the word gadgets
7) There is a rogue sentence at the end of page 3 or if it is meant to be a heading it needs a style
8) There is an inconsistent use of styles for sub headings in section 3
9) There is a repeated quote in the theme 2 section and the theme 3 heading is repeated
10) The headings in section 4 are not consistent with the introduction in section 4 but also earlier in the paper. Try to be consistent on the key points/findings of the paper
11) In section 4.1 there is the word table, it should be tablet
12) Section 4.4 and 5 are too short - so the interesting take aways from this paper are not clear.
Finally I do believe the Abstract could be more research focused and compelling.
Example of a stronger Abstract
Information, Communication and Technology (ICT) has become an indispensable tool in education, particularly during the COVID-19 pandemic. However, integrating ICT into teaching and learning has been a daunting challenge in many developing countries such as Uganda. This qualitative case study investigated the challenges faced by primary school teachers in implementing ICT in teaching Science, Technology, Engineering, and Mathematics (STEM) subjects in Uganda. The study found that teachers face various challenges in integrating ICT into the curriculum. These challenges were categorized into three themes: infrastructure and internet connectivity, individual factors and administrative support, and curriculum and learner support materials. A significant obstacle was the lack of access to computers, internet connectivity, and ICT textbooks. Additionally, teachers lacked the necessary ICT skills to integrate technology into their teaching methods, and administrative support was insufficient. The study's findings have significant implications for the effective integration of ICT into teaching and learning in Uganda's primary schools. Addressing the challenges identified in this study will require a multi-pronged approach that includes increasing investment in infrastructure, providing training and support to teachers, and developing relevant and appropriate ICT curriculum materials. The study's recommendations can inform policymakers and stakeholders in Uganda and other developing countries seeking to integrate ICT into their education systems. In conclusion, this study highlights the challenges that primary school teachers face in implementing ICT in teaching STEM subjects in Uganda. By addressing the identified challenges, policymakers and stakeholders can take steps towards improving ICT integration in primary school education and bridging the digital divide in developing countries.
Example of a stronger conclusion by combining section 4.4. and section 5 and some information from the earlier sections.
This research study aimed to investigate the challenges faced by primary school teachers in integrating Information and Communication Technology (ICT) in teaching Science, Technology, Engineering, and Mathematics (STEM) subjects during the post-COVID-19 period. The study focused on two private Seventh Day Adventist primary schools in Uganda, representing urban and rural contexts. The study explored the following research questions: What ICT gadgets do teachers use in teaching and management? How does the school support teachers to integrate ICT in teaching STEM subjects? What challenges do teachers face in integrating ICT in primary school STEM subjects, and how do they deal with them?
The study identified three main challenges: infrastructure and internet connectivity, individual factors and administrative support, and curriculum and learner support materials. The lack of adequate infrastructure and internet connectivity was a significant challenge for both rural and urban schools. Although the urban school had a 3:1 ratio of computers to learners [45], this is unlikely to be representative of the majority of primary schools in sub-Saharan Africa. The study found that the challenges faced by teachers in integrating ICT had significant implications for meeting the objectives of the National Curriculum to improve learners' interest and participation in STEM subjects.
The study also revealed a digital divide between rural and urban schools at individual, societal, and geographical levels. Even with the necessary infrastructure, some teachers showed resistance to ICT integration, which affects their attitudes towards technology in the classroom [4], [55]. Additionally, expensive data was a significant barrier to integrating ICT in teaching STEM subjects, which may limit learners' development of the 5th Industrial Revolution (5IR) skills necessary for a technology-based society [49].
This study makes a significant contribution to the literature on challenges in integrating ICT in STEM subjects in primary schools post-COVID-19, especially in private Seventh Day Adventist primary schools in Uganda. The findings highlight the urgent need to address the digital divide in Uganda and provide adequate infrastructure, administrative support, and teacher training to integrate ICT in teaching STEM subjects. It is recommended that policymakers and school administrators take appropriate measures to improve ICT infrastructure, reduce the cost of data, provide administrative support and teacher training to bridge the digital divide, and ensure learners develop the necessary skills for the 5IR.
IMPORTANT: Note I have used ChatGPT using your original paragraphs to create these more compelling paragraphs.

Author Response
My responses to the reviewers' 1 and 2 comments are attached as one file.

Reviewer 2 Report
Thank you for inviting me to be a reviewer of the manuscript entitled Primary teachers’ challenges in implementing ICT in Science Technology Engineering and Mathematics (STEM) in the post-pandemic era in Uganda. This document is really impressive in terms of your efforts to demonstrate the power of your study.
Introduction: In the introductory part of the study, the basic ideas and directions of the entire study are presented. 30 quality references are used in the introductory part. The research questions are presented at the end of this section. I propose to place these research questions in the next part of the study.
Research: The aim of the study was to investigate the problems of primary school teachers in the integration of ICT in the teaching of science subjects in the period after COVID-19. Research proposal and design are appropriately used in this study. The chosen description of the research qualitative inquiry is not very detailed. I recommend providing more detailed information. I consider the research sample to be small for quality research. In the research, I propose to add schemes for solving the research investigation, as well as for displaying the data analysis. This would complete the findings. The findings of the research team are interesting and form the basis for further follow-up research, which I would appreciate in this or a follow-up study. I recommend adding schemas from the axial coding of topics and subtopics. I recommend adding a chat structure.
Discussion and conclusion: In this section, the authors qualitatively discuss their findings and results of the research investigation using a questionnaire. The authors refer to other works and try to compare the results. The entire described part of the study is quite comprehensive, but not boring. I recommend expanding the conclusion section on the future direction of research.
References: This study refers to 56 scientific references, sources and publications. The references used are up-to-date and of sufficient quality and are a suitable theoretical basis for this study. I recommend fixing and unifying the formatting of the references.
Conclusion: This study represents a contribution to this area of research. The basic ideas of the submitted manuscript are interesting. I see great potential in the study for further follow-up research. I would recommend partial changes and modifications in the research part of this study. However, some passages of the study are very descriptive and lengthy. This is sometimes confusing. Therefore, I would suggest shortening and simplifying them.
Author Response
My responses are attached along with reviewer 1

Round 2
Reviewer 1 Report
This is much improved and almost ready for publication. There are some layout issues with the two figures and the appendix. They could both be improved to look a bit more professional.
There are also some paragraph style formatting issues (sometime single spacing, sometime double, sometimes dual-justified, sometimes just left-justified.
The content of the paper is now great, though would benefit from an English Language/Expression proof read
Reviewer 2 Report
The proposed changes correspond and the article can be published.